# Multiplex Ligation Probe Amplification and Sanger Sequencing: Light and Shade in the Diagnosis of Lysosomal Storage Disorders

**DOI:** 10.3390/biomedicines13040973

**Published:** 2025-04-16

**Authors:** Martina Vinci, Carmela Zizzo, Marta Moschetti, Miriam Giacomarra, Monia Anania, Giulia Duro, Tiziana Di Chiara, Maria Russo, Elisa Messina, Paolo Colomba, Giovanni Duro

**Affiliations:** 1Institute for Biomedical Research and Innovation (IRIB), National Research Council (CNR), 90146 Palermo, Italy; martina.vinci@irib.cnr.it (M.V.); carmela.zizzo@irib.cnr.it (C.Z.); marta.moschetti@irib.cnr.it (M.M.); miriam.giacomarra@irib.cnr.it (M.G.); monia.anania@irib.cnr.it (M.A.); maria.russo@irib.cnr.it (M.R.); elisa.messina@irib.cnr.it (E.M.); giovanni.duro@irib.cnr.it (G.D.); 2Internal Medicine, Ospedale Cattinara, 34149 Trieste, Italy; giulia.duro@libero.it; 3Excellence Department of Health Promotion, Maternal and Child, Internal and Specialist Medicine “G. D’Alessandro”, University of Palermo, 90127 Palermo, Italy; tiziana.dichiara@unipa.it

**Keywords:** MLPA, lysosomal storage disease, Fabry disease, Gaucher disease, Pompe disease

## Abstract

**Background:** Multiplex Ligation Probe Amplification (MLPA) is a widely used technique for the diagnosis of lysosomal storage diseases (LSDs). It analyses over 40 DNA sequences in a single reaction, identifying copy number variations and large deletions/insertions in genes. The diagnostic process in LSDs starts with analysis of the missing or reduced enzyme, followed by genetic investigation and, if possible, a search for accumulated substrates. However, while genetic analysis using Sanger sequencing is excellent at detecting small genetic variations such as single-nucleotide variants (SNVs) and small insertions or deletions, it cannot detect large deletions or insertions. **Methods:** In the present study, a total of 800 patients with clinical suspicion of Fabry, Gaucher, or Pompe diseases were investigated. An enzyme assay was carried out on each patient, followed by genetic analysis using PCR, Sanger sequencing, and MLPA. **Results:** Nine patients with deficient or absent enzyme activity had Sanger sequencing results that could not confirm the molecular genetic diagnosis because either no mutation (Fabry) or only one mutation (Gaucher and Pompe) was identified. Subsequent analysis by MLPA identified two males with a hemizygous deletion and two females with a heterozygous deletion for FD. For PD, one female and two males had a heterozygous deletion. For GD, one male had a homozygous deletion and one female had a heterozygous deletion. The remaining patients were analyzed by MLPA with negative results. **Conclusions:** The results obtained suggest that MLPA should be used in combination with classical sequencing methods to ensure a correct and timely diagnosis of LSDs.

## 1. Introduction

Although most inherited diseases are caused by point mutations, small deletions, or insertions in the DNA sequence of specific genes, large deletions or insertions spanning one or more exons are also frequently implicated. Consequently, accurate characterization of these variations is a crucial step in confirming the clinical diagnosis and studying the genotype-phenotype correlation [1]. Furthermore, the genetic basis of many diseases has been shown to be related to copy number variation (CNV), defined as a DNA segment longer than 1 Kb, with a variable copy number compared to the reference genome [2]. As neither conventional cytogenetic analysis nor classical DNA sequencing can detect exon-spanning deletions/insertions or CNV, it is essential to establish suitable approaches for the study of these particular mutations. Among the various approaches that have emerged in recent years, Multiplex Ligation Probe Amplification (MLPA) has received significant attention due to its ability to analyze more than 40 DNA sequences in a single reaction, facilitating the detection of CNV in specific genes, including small intragenic rearrangements, deletions, or duplications [3]. The development of MLPA was to confirm or exclude gross gene rearrangements in several monogenic diseases [4], with more than 300 probe sets commercially available from MRC Holland [3].

### 1.1. Principles of the MLPA Technique

MLPA is a technique that utilizes the principle of multiplex PCR, employing over 40 probes that are specific to diverse DNA sequences, predominantly the exons of the gene of interest. This approach enables the assessment of the relative copy number of each gene, in addition to the identification of deletions or insertions. The MLPA technique is subdivided into six stages: (1) DNA denaturation; (2) probe hybridization; (3) ligation of the hybridized probes; (4) amplification of the hybridized probes with common primers; (5) separation of the amplification products by electrophoresis; and (6) quantification of amplification products. In the first two steps, the DNA is denatured and incubated with the probes specific to the gene of interest. Each MLPA probe is composed of two oligonucleotides that are specifically designed to recognize adjacent target sequences with no nucleotides between them. Only in the presence of a perfect pairing, after the hybridization reaction, can these probes ligate into a single probe and undergo amplification using primers that recognize common non-binding tail sequences at the 5’ ends of the probes. Each amplification reaction is performed using two primers, only one of which is fluorescently labelled. Following this, the PCR products are subjected to capillary electrophoresis under denaturing conditions and visualized by means of an electrophoretogram. The relative height of the peak area is then measured, which quantifies the amount of PCR product compared to the control DNA samples, thus indicating the relative amount of the target DNA sequence in the DNA sample [3,5]. The quality of the reaction is then determined by the presence of the control peaks, which provide information regarding the efficiency of amplification and the correct amount of DNA used for the reaction. A crucial point in using MLPA as a genetic test for the molecular diagnosis of gene deletions/duplications is the interpretation of the results. In the case of deletions in the region of interest, the ligation reaction does not take place, resulting in a reduction/absence of the amplicon, detectable as a reduction/absence of the corresponding peak. Conversely, in the context of duplications, an increase in the amplicon is observed, resulting in a higher peak with a larger area (Figure 1) [3]. Consequently, various analysis strategies have been developed to enable the correct interpretation of the data. Of the various software, the most widely used is Coffalyser, an Excel-based program capable of performing data normalization and signal correction [6,7,8]. Despite the high reproducibility of amplification reactions utilizing MLPA, variability in amplification efficiency among different target sequences is observed. Therefore, the analysis of a single reaction profile is insufficient for obtaining meaningful information. Furthermore, each SALSA MLPA kit contains internal controls that can highlight errors or anomalies that occurred during the experimental steps [3].

MLPA is a simple and innovative technique for determining the number of copies of a DNA sequence. Its advantages include: (1) the ability to scan up to 40 loci per gene; (2) a wide range of applications; (3) high reproducibility and ease of execution; (4) the ability to discriminate sequences differing by a single nucleotide; and (5) the ability to discriminate small variations in copy number (three versus two) of a gene in a complex mixture [3]. Since MLPA is highly sensitive to the type of sample used for DNA extraction (blood or DBS), it is recommended that comparative analyses be performed using DNA samples extracted from the same tissue and by the same method. MLPA and Sanger sequencing focused on the coding sequence (exons) may fail to detect certain rare mutations present in the regulatory regions of a gene or that are caused by a structural rearrangement, e.g., an inversion or balanced translocation. In these cases, genome sequencing, gene expression, fluorescence in situ hybridization, or other methods may be required.

MLPA can be used in the diagnosis of various lysosomal storage diseases (LSDs), which are caused by genetic mutations that alter the activity of lysosomal enzymes. These enzymes are essential for the digestion of various substrates. In the absence of a functional enzyme, substrate that would typically be subject to degradation accumulates, resulting in cellular and organ damage. The prognosis depends on the type of substrate and the severity of the symptoms. Some diseases have a severe and often fatal prognosis in the first years of life, while others, such as Gaucher, Fabry, and Pompe diseases, can be managed with specific treatments.

Since 2019, the Centre for Research and Diagnosis of Lysosomal Accumulation Diseases of the IRIB-CNR in Palermo has been involved in the study of several LSDs, including Anderson–Fabry disease, Gaucher disease, and Pompe disease.

### 1.2. MLPA in the Diagnosis of Fabry, Gaucher, and Pompe Diseases

Fabry disease (FD OMIM #301500) is a lysosomal storage disorder (LSD) caused by a deficiency in the lysosomal hydrolase α-galactosidase A (α-GAL A). This results in an accumulation of sphingolipids (predominantly globotrioasylceramide) containing terminal α-galactose residues within the lysosomes of many organs and tissues, including the kidneys, arteries, and heart [4,9]. The *GLA* gene, which is responsible for encoding the *α-GAL A*, is located on the long arm of the X chromosome, thus causing an X-linked disease [10]. Over 1000 mutations have been identified in this gene, including missense, nonsense, splicing, deletion, and insertion mutations [11,12]. In men with a clinical diagnosis of FD, routine PCR analyses can easily identify the presence of deletions, given the hemizygosity of the X chromosome. Such analyses are not always reliable in females, due to the presence of the two gene copies. In the case of a large encompassing one or more exons deletion in one of the two chromosomes, this may not be detected using the classic Sanger sequencing technique. Consequently, in the presence of suggestive symptoms, it is imperative to supplement routine sequencing analyses with orthogonal tests, such as MLPA, to reveal possible genetic alterations [4,9,13,14]. The employment of the MLPA technique in Fabry disease is therefore necessary to ensure a correct diagnosis, thereby reducing the risk of false negatives [15].

LSDs encompass a range of pathologies, including Gaucher disease (GD OMIM #230800) and Pompe disease (PD OMIM #232300), that are characterized by autosomal recessive transmission.

Gaucher disease (GD) is caused by mutations in the *GBA1* gene, which codes for the lysosomal enzyme glucocerebrosidase (GCase), also known as acid β-glucosidase. This deficiency of GCase activity leads to an accumulation of its substrate, glucosylceramide (GlcCer), in the lysosomes of macrophages, resulting in their enlargement and subsequent designation as ‘Gaucher cells’. These cells primarily infiltrate the bone marrow, spleen, and liver, contributing to the manifestations of the disease [16]. The *GBA* gene, which maps to chromosome 1, exhibits 95% homology with a pseudogene (*GBAP1*) located 15 kb downstream [17]. Consequently, the chromosomal region is subject to reciprocal and non-reciprocal homologous recombination events, resulting in deletions, duplications, and rearrangements of the gene-pseudogene complex, which is responsible for the disease [18].

Pompe disease (PD), also known as glycogenosis type II or acid maltase deficiency, is a chronic neuromuscular disease caused by mutations in the *GAA* gene, located on chromosome 17, that lead to a reduction in the activity of the acid alpha-glucosidase enzyme (*GAA* enzyme). This results in an accumulation of glycogen within lysosomes, which damages muscles in many organs [19,20]. The classification of PD is based on three main factors: the age of onset, the organs involved, and the severity of the condition. Two distinct forms of PD have been identified: the classic infantile-onset form (IOPD), characterized by severe muscle weakness and hypertrophic cardiomyopathy, and the non-classic late-onset form (LOPD), which is characterized by slowly progressive muscle weakness. The mutational spectrum in the *GAA* gene is characterized by significant heterogeneity, encompassing point mutations, as well as small and large deletions and insertions [19]. In contrast to Fabry disease, Gaucher and Pompe diseases are autosomal recessive disorders that require a mutation in both the maternal and paternal alleles. It is important to note that deletions or insertions, copy number variations, and complex gene-pseudogene rearrangements (in the case of Gaucher disease) can often present significant challenges during mutation analysis [18]. In both diseases, the MLPA technique has the ability to simultaneously analyze multiple regions of the genes of interest for copy number variations, allowing for a more complete and accurate diagnosis and reducing the risk of false negatives.

In the present study, an approach based on MLPA (MRC-Holland, Amsterdam, the Netherlands) was established, which allowed the identification of specific homozygous and heterozygous deletions in nine patients with Fabry, Gaucher, and Pompe disease. In this context, MLPA has been utilized to identify mutations that were not detected by canonical Sanger sequencing in patients exhibiting pathological or borderline enzyme activity.

A unique kit (SALSA MLPA Probe-mix) was utilized for each disease, with the specific probe for each gene of interest: ‘P453-A2 *GAA*’ for Pompe disease (20 exons-18 probes), ‘P338-B2 *GBA*’ for Gaucher disease (11 exons-8 probes), and ‘P159-A5 *GLA*’ (7 exons-8 probes) for Fabry disease. The results obtained from this analysis allowed for the completion of a mutational analysis, thereby highlighting the importance of this technique in diseases whose pathogenesis is related to the presence of deletions and/or duplications of specific genes.

## 2. Materials and Methods

### 2.1. Patients

Peripheral blood was collected using ethylenediaminetetraacetic acid (EDTA) as an anticoagulant, following which it was dried on absorbent paper (dried blood spot, DBS). Genetic and enzymatic studies were performed at the Centre for Research and Diagnosis of Lysosomal Storage Disorders of IRIB-CNR in Palermo and were approved by the Hospital Ethics Committee of the University of Palermo. Signed informed consent was obtained from patients.

### 2.2. Enzyme Activity Assays

Enzyme activities were calculated using a fluorimetric dried-blood spot (DBS) assay, described by Chamoles et al. [21], with some modifications [unpublished data]. In summary, a spot containing 10 µL of blood in a 6 mm diameter paper circle was placed in a 96-well plate, and following an incubation period of 18 h at 37 °C (900 RPM), the reaction was stopped with 250 µL of 0.1 mol/L ethylenediamine (pH 11.4). It should be noted that a different synthetic substrate was utilized for each disease. Subsequent to the completion of the reaction, the molecules were analyzed for fluorescence using a fluorometer set at the 4-methylumbelliferone (4MU) wavelength (365–488 nm). Finally, the fluorescence values obtained were related to a standard curve constructed with increasing concentrations of 4MU. The values considered normal for each disease were as follows: ≥3 nmol/L/h for Fabry disease; ≥6 nmol/L/h for Pompe disease; and ≥2.5 nmol/L/h for Gaucher disease.

### 2.3. Genetic Analysis

Genomic DNA was isolated from dried blood spots using silica-coated magnetic particles in an automated nucleic acid purification extractor (EZ1&2 DNA Investigator Kit, Qiagen, Hilden, Germany). The concentration of DNA was measured using a biophotometer (D30, Eppendorf, Hamburg, Germany). The search for mutations in the *GLA*, *GAA*, and *GBA1* genes was conducted using Sanger sequencing, and the PCR products were purified and sequenced at Eurofins Genomics (Ebersberg, Germany).

### 2.4. Biomarkers Assay

The determination of Lyso-Gb3 and Lyso-Gb1 in blood was achieved through the utilization of tandem mass spectrometry (MS/MS), a method previously delineated by Polo et al. [22].

### 2.5. MLPA Analyses

We conducted an MLPA analysis on DNA samples to identify major rearrangements of the *GLA*, *GAA*, and *GBA* genes, using the SALSA MLPA kits P159-A5 *GLA*, P453-A2 *GAA*, and P338-B2 *GBA*, in accordance with the manufacturer’s instructions (MRC Holland).

### 2.6. Data Analysis

Data analysis is performed using the Coffalyser.Net software. After within-sample and between-sample normalization, the dosage quotient (DQ) is calculated using Excel worksheets.

## 3. Results

This study of 800 patients included 409 with a clinical suspicion of Fabry disease (FD), 233 with a clinical suspicion of Gaucher disease (GD), and 158 with a clinical suspicion of Pompe disease (PD) (Figure 2). These 800 patients with signs and symptoms attributable to one of the three diseases were subjected to biochemical analysis. The test detected reduced or absent enzyme activity. Subsequently, genetic analysis using Sanger sequencing either did not detect mutations in specific genes (or only one causative mutation was found). A more in-depth study was then conducted using MLPA, which identified nine patients. For FD, two unrelated males with a hemizygous deletion of exons 3 and 4, and their mothers with the same heterozygous deletion; for PD, a female with a heterozygous deletion of exons 9 and 18 and two unrelated males with a heterozygous deletion of exon 18; finally, for GD, a male with a homozygous deletion of exon 10 and a female with a heterozygous deletion of exon 6. MLPA was negative in the remaining patients.

### 3.1. Patients with Clinical Suspicion of Fabry Disease

Cases 1 and 2 are two unrelated male patients, aged 32 and 46 years, respectively, with signs and symptoms attributable to Fabry disease: acroparesthesias, cornea verticillata, angiokeratomas, and LVH (Left Ventricular Hypertrophy) (Table 1). Enzymatic analysis of *α-GAL A* revealed little to no enzymatic activity in both subjects (0 and 0.2 nmol/mL/h) (Table 1). However, genetic analysis of the *GLA* gene, performed using PCR and Sanger sequencing, did not allow for the amplification of some exons. Consequently, an in-depth study was conducted utilizing the MLPA technique, which revealed the complete absence of the peaks corresponding to the probes for exons 3 and 4, thus confirming their hemizygous deletion, given that the subjects were male (Figure 3 and Figure 4). Where possible, we also performed the determination of Lyso-Gb3, the accumulation substrate in Fabry disease, which was elevated (36.64 and 35.37 nmol/L) compared to the normal reference range (<2.3 nmol/L), confirming the diagnosis of the disease (Table 1). The study was also extended to family members, leading to the identification of two additional patients, the mothers of the respective probands. Both women, aged 64 and 70, had high Lyso-Gb3 values (5.47 and 9.67 nmol/L) and enzyme activity slightly above the reference value (3.5 and 9.2 nmol/mL/h) (Table 1). In the women, the enzyme activity values are not informative and still presuppose genetic investigation, which did not detect any mutations, even though both had symptoms attributable to Fabry disease (LVH, rheumatoid arthritis) (Table 1). The findings necessitated further investigation by MLPA, which revealed a 50% reduction in the height of the peaks corresponding to the probes for exons 3 and 4, confirming the presence of a heterozygous deletion in both subjects (Figure 5 and Figure 6).

### 3.2. Patients with Clinical Suspicion of Pompe Disease

The following three patients had a clinical diagnostic suspicion of Pompe disease, which was then investigated using an enzymatic and genetic study of acid alpha-glucosidase, also known as acid β-maltase (Table 2).

Case 3 is a one-month-old girl suffering from the classic infantile form of the disease (IOPD) with onset in the first few weeks of life, with severe cardiac involvement, generalized hypotonia, muscle weakness, proteinuria, and poor growth (Table 2).

The enzymatic activity of acid β-maltase was absent (0.9 nmol/h/mL), although the genetic study performed by Sanger sequencing only identified a single heterozygous mutation (c.784 G>A: p.E262K) in exon 4 of the *GAA* gene (Table 2), not accounting for the absence of activity of the enzyme. An in-depth study was then conducted using MLPA to search for large deletions/insertions. This study revealed that there was a halving of the peaks corresponding to specific probes for exons 9 and 18 of the *GAA* gene. This finding confirmed a heterozygous deletion that was specific to these exons (Figure 7). The study was expanded to the proband’s family members, who were asymptomatic with normal enzyme activity (Table 2). In the mother, by MLPA, we identified the deletion of exons 9 and 18 of the *GAA* gene in heterozygosity (Table 2), while in the father, it was negative. Finally, in the father, the mutation c.784 G>A: p.E262K was identified in heterozygosity using Sanger sequencing (Table 2), confirming compound heterozygosity in the girl and the carrier status of both parents.

Cases 4 and 5 are two unrelated males aged 9 and 24, respectively, suffering from the late-onset form of the disease (LOPD) that manifested in both with hyperCKemia (increased creatine phosphokinase (CPK)), increased AST/ALT, increased LDH, and mild neck weakness (Table 2). In both cases, the reduced enzymatic activity (1.5 and 3.3 nmol/h/mL) required genetic study by Sanger sequencing, which showed only one causative mutation in heterozygosity (c.-32-13 T > G) in intron 1 of the *GAA* gene, not explaining the pathological enzymatic activity (Table 2). Consequently, we conducted an additional study utilizing the MLPA method, which revealed a halving of the peak corresponding to the specific probe for exon 18 of the *GAA* gene. These results allowed us to confirm a heterozygous deletion specific to this exon (Figure 8 and Figure 9). In both cases, it was not possible to extend the study to the probands’ relatives. However, the reduced enzymatic activity and the clinical features strongly suggested that the two mutations were biallelic.

### 3.3. Patients with Clinical Suspicion of Gaucher Disease

Case 6 is a 2-year-old male patient who exhibited symptoms that were strongly attributable to Gaucher disease, including hepatosplenomegaly, anemia, thrombocytopenia, and gastroesophageal reflux (Table 3). The enzymatic study of acid β-glucosidase, conducted within the laboratory, identified the absence of enzyme activity (0.4 nmol/h/mL) (Table 3). Furthermore, a genetic study employing Sanger sequencing failed to identify any mutations in the *GBA1* gene, thus not explaining the absence of enzyme activity. After this, the study was expanded by MLPA, which revealed the absence of the peak corresponding to the probe specific to exon 10 of the *GBA1* gene, thus confirming a homozygous deletion specific to this exon (Figure 10). The MLPA study was also extended to the proband’s asymptomatic family members with normal enzyme activity (Table 3) and allowed the deletion of exon 10 in heterozygosity to be confirmed in both mother and father (Table 3). Consequently, an exhaustive investigation of exon 10 of the *GBA1* gene was undertaken, leading to the realization that the exon 10 deletion observed in patient 6 had been caused by a rearrangement with the *GBA1* pseudogene.

Case 7 was a 22-year-old female patient who exhibited signs and symptoms indicative of Gaucher disease, including hepatomegaly, splenomegaly, and thrombocytopenia (Table 3). The enzymatic activity of acid β-glucosidase was absent (0.5 nmol/h/mL) (Table 3). In addition, the study of Lyso-Gb1, a product of accumulation in Gaucher disease, revealed a pathological value (498.0 ng/mL) compared to the normal reference range (≤6.8 ng/mL) (Table 3). The genetic study performed in Sanger sequencing allowed the identification of a single homozygous mutation in exon 9 of the *GBA1* gene (c.1226 A>G) (Table 3), which codes for the p. N409S mutation, historically known in the literature as N370S [23]. The study was also extended to the proband’s asymptomatic family members with normal enzyme activity (Table 3). The genetic analysis performed by Sanger sequencing identified a heterozygous mutation in exon 9 of the *GBA1* gene c.1226 A>G: p. N370S in the father (Table 3); while in the mother, the genetic analysis performed by Sanger sequencing did not find a mutation. Subsequent investigation utilizing the MLPA method, revealed a 50% reduction in the peak corresponding to the probe of exon 6 of the *GBA1* gene, thereby indicating the presence of a heterozygous deletion in the mother (Figure 11). MLPA analysis was also performed in the proband, confirming the deletion in heterozygosity of exon 6 (Figure 12). Therefore, the N370S mutation found in the proband in Sanger sequencing was only due to the contribution of the paternal allele, because the other allele was deleted in the amplified region [24]. These results thus confirmed the presence of compound heterozygosity in the patient and the carrier status of both parents.

## 4. Discussion

To ensure a correct and timely diagnosis in the presence of a clinical diagnosis of LSDs, the use of MLPA has increased considerably over the years. The first diagnostic step is the testing of enzyme activity, which is significantly reduced or absent in affected patients. The second diagnostic step involves molecular testing, utilizing PCR and Sanger sequencing techniques, which serves to confirm the enzyme tests and identify specific mutations in the genes of interest [25]. Where necessary, substrate accumulation in the blood, urine, or tissue is also assessed. However, it should be noted that molecular analysis by Sanger sequencing may yield a negative or uncertain result due to the presence of small intragenic rearrangements and deletions/insertions.

For several years, our laboratory, specializing in the research and diagnosis of lysosomal storage diseases, has been using an MLPA-based approach to improve the clinical diagnosis of Fabry, Gaucher, and Pompe diseases. Specifically, this study has demonstrated the efficacy of this technique in detecting large deletions/insertions in subjects with pathological or borderline enzyme activity, where classic Sanger sequencing has failed.

Fabry disease, an X-linked type, manifests itself in a different manner in men and women; consequently, the diagnostic approach also varies. In male subjects diagnosed with FD, the assessment of enzyme activity is crucial and should be followed by genetic analysis in cases of enzyme alterations. The initial two cases presented were unrelated males exhibiting symptoms consistent with the disease. However, despite the absence of enzyme activity and elevated Lyso-Gb3 levels (accumulation of substrate in Fabry disease), genetic analysis using Sanger sequencing failed to amplify specific exons. Therefore, it was essential to perform MLPA analysis, which revealed a hemizygous deletion in exons 3 and 4 in both subjects, given the single copy of the gene. Sanger sequencing is not designed to detect large-scale structural variations, such as large deletions, as its focus is on point mutations or small insertions/deletions [26]. In the two case studies presented here, the hemizygous deletion in question resulted in the elimination of the binding region of the primers, thereby preventing the visualization of the bands in the exons of interest. The diagnostic approach for female subjects with FD is different. In contrast to male subjects, the primary diagnostic test for female subjects is genetic analysis, as enzyme activity is not indicative in women due to the lyonization phenomenon [10]. The focus then shifted to the mothers of the respective probands. The women exhibited symptoms consistent with the disease, but the genetic analysis performed using Sanger sequencing did not reveal any mutations. The female subjects in this study exhibit cellular mosaicism, comprising cells that express the X chromosome with the deletion and a mixed population of cells that express the normal X chromosome. Consequently, the effect of the deletion may be masked by the presence of the normal X chromosome. Consequently, we conducted a more extensive investigation using the MLPA technique, which revealed a heterozygous deletion in exons 3 and 4, given the presence of the wild-type copy of the gene. In this case, it is important to underline that without the family study and the use of the MLPA technique, we would not have detected the presence of mutations in the two women, considering the double X chromosome [4].

In contrast to Fabry disease, genetic analysis for Gaucher and Pompe diseases, which are autosomal recessive, is only conducted following an assessment of enzyme activity. The manifestation of the clinical phenotype in autosomal recessive diseases is contingent upon the presence of bi-allelic mutations (of paternal and maternal origin).

For Pompe disease, we have documented three cases of unrelated patients who exhibited minimal enzyme activity and classical symptoms. In the third case, genetic analysis using Sanger sequencing revealed a single mutation in heterozygosity in exon 4. Later, an in-depth study using MLPA revealed a heterozygous deletion in exons 9 and 18. The studies were also extended to the proband’s family members, and the genetic analysis carried out using Sanger sequencing had detected the mutation in heterozygosity in the father, while it was negative in the mother. Subsequent analysis using the MLPA method identified the deletion in exons 9 and 18 in the mother, thus confirming the carrier status of both parents. In cases 4 and 5, genetic analysis performed by Sanger sequencing had also revealed the presence of a single heterozygous mutation in exon 1. Consequently, MLPA analysis conducted on both patients revealed a heterozygous deletion in exon 18 of the gene. As it was not possible to extend the genetic investigation to the probands’ family members, the severe symptoms (including glycogen accumulation in case 5) associated with reduced enzyme activity suggested that the two mutations were found in both alleles.

For Gaucher disease, two cases were reported, of which one child exhibited typical symptoms and absence of enzyme activity. Despite this, the genetic analysis performed through Sanger sequencing had not detected any mutation. Only later did an in-depth study using MLPA identify a homozygous deletion of exon 10. The study was expanded to include the proband’s asymptomatic relatives, and the MLPA analysis revealed a heterozygous deletion of exon 10 in both parents. Subsequent studies utilizing Sanger sequencing analysis revealed that the deletion of exon 10 in the proband had been caused by a rearrangement with the *GBAP1* pseudogene. It was observed that the *GBAP1* pseudogene exhibits a 95% homology with the *GBA1* gene, consequently resulting in reciprocal and non-reciprocal homologous recombination events, which in turn are responsible for the mutations that underpin the disease [18]. Finally, the last case was a girl with symptoms attributable to the disease, absent enzyme activity, and an elevated Lyso-Gb1 value (accumulation product in Gaucher disease). Initially, genetic analysis using Sanger sequencing had detected a homozygous mutation in exon 9 of the gene. In accordance with the protocol for autosomal recessive diseases, the study was expanded to the proband’s parents. While the genetic analysis using Sanger sequencing found the same mutation in the father, it was negative in the mother, not explaining the homozygosis condition found in the girl initially. Consequently, the MLPA analysis conducted on the mother revealed a heterozygous deletion in exon 6, analogous to that observed in the proband. This finding elucidated the initial result obtained from the Sanger sequencing analysis, which indicated apparent homozygosity due to the exclusive contribution of the paternal allele. This was attributed to the absence of amplification of the maternal allele, which contained a deletion in exon 6, within the region spanning exon 5 to intron 11 [27].

By comparing our results with other work in the literature using MLPA in conjunction with other diagnostic techniques, such as clinical exome sequencing (CES) to study GD [28] or other work on FD and PD [29,30], our study allows us to highlight the importance of this technique to confirming the molecular genetic diagnosis of LSDs and also gives an idea of the frequency of patients found in these years: for FD 1:100, for PD 1:50, and for GD 1:115.

## 5. Conclusions

The present study confirms that in subjects with large deletions, a molecular genetic diagnosis of an LSD may not be possible if only the Sanger sequencing method is used.

Consequently, we propose that in cases of LSDs characterized by autosomal recessive transmission, MLPA should be conducted in all instances where the presence of clinical symptoms and no or significantly reduced enzyme activity are not fully explained by Sanger sequencing, or where familial testing does not align with the results obtained by Sanger sequencing, e.g., in the case of apparently homozygous mutation being due to a heterozygous deletion and a second mutation. In cases of LSDs characterized by X-linked transmission, MLPA should be performed in all suspected females as a first-line test alongside Sanger sequencing, as biochemical testing may be normal due to lyonization. In males with pathological enzyme activity and clinical symptoms of an X-linked disorder and uninformative Sanger sequencing results, MLPA is essential to detect the presence of a large deletion or insertion. In conclusion, our data suggest that the MLPA technique may improve the genetic analysis of LSDs, limiting possible delays in molecular genetic diagnosis.

## Figures and Tables

**Figure 1 biomedicines-13-00973-f001:**
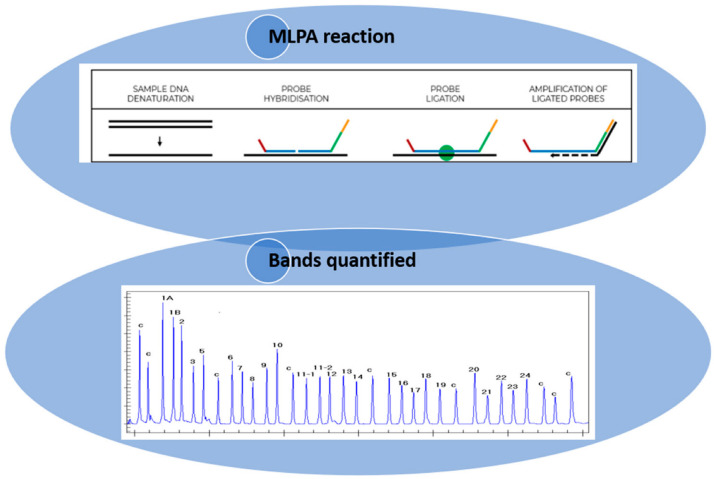
Illustration of how the MLPA technique is performed (https://support.mrcholland.com/kb/articles/getting-started-with-mlpa accessed on 13 March 2025) and how the bands are quantified. Each peak is the amplification product of a specific probe and indicates a difference in peak height or peak area relative to a control.

**Figure 2 biomedicines-13-00973-f002:**
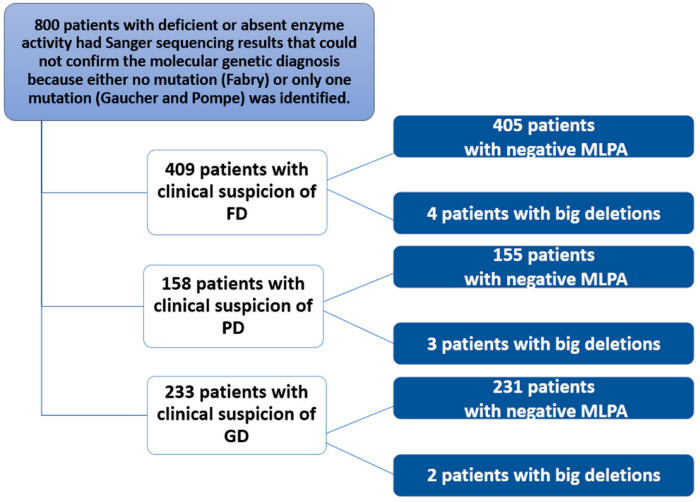
Flowchart of the study conducted in 800 patients with clinical suspicion of Fabry, Gaucher, or Pompe disease.

**Figure 3 biomedicines-13-00973-f003:**
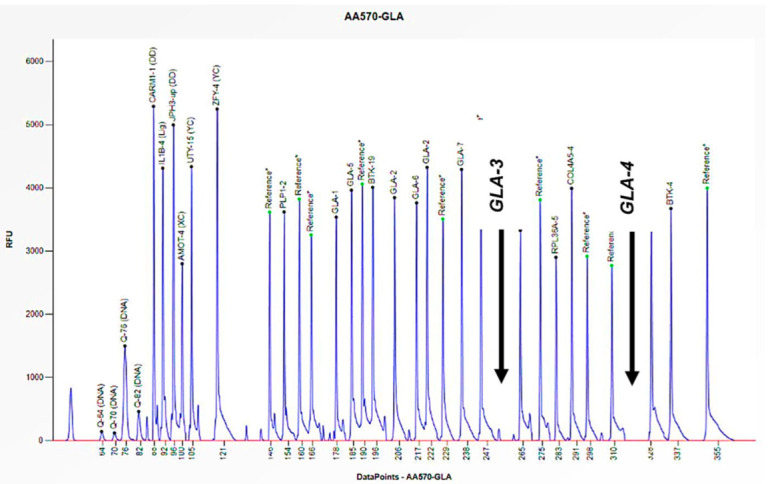
The electrophoretogram of the male patient identified with the code AA570 demonstrates a complete absence of the peak corresponding to the probes for exons 3 and 4 of the *GLA* gene in comparison with the reference probe, thus indicating the presence of a hemizygous deletion.

**Figure 4 biomedicines-13-00973-f004:**
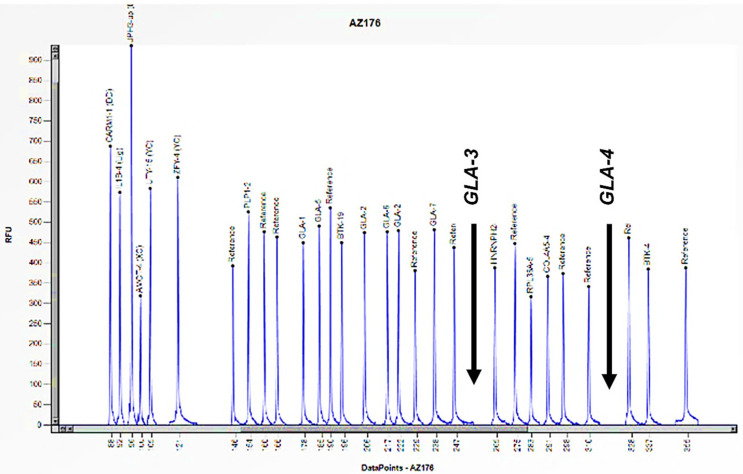
The electrophoretogram of the male patient identified with the code AZ176 demonstrates a complete absence of the peak corresponding to the probes for exons 3 and 4 of the *GLA* gene in comparison with the reference probe, thus indicating the presence of a hemizygous deletion.

**Figure 5 biomedicines-13-00973-f005:**
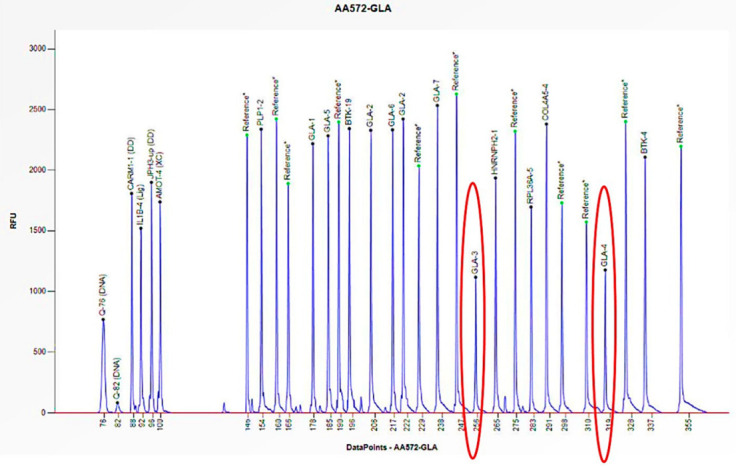
The electrophoretogram of the female patient with the code AA572 demonstrates a 50% reduction in the peak corresponding to the probes for exons 3 and 4 of the *GLA* gene in comparison to the reference probe, thus indicating the presence of a heterozygous deletion.

**Figure 6 biomedicines-13-00973-f006:**
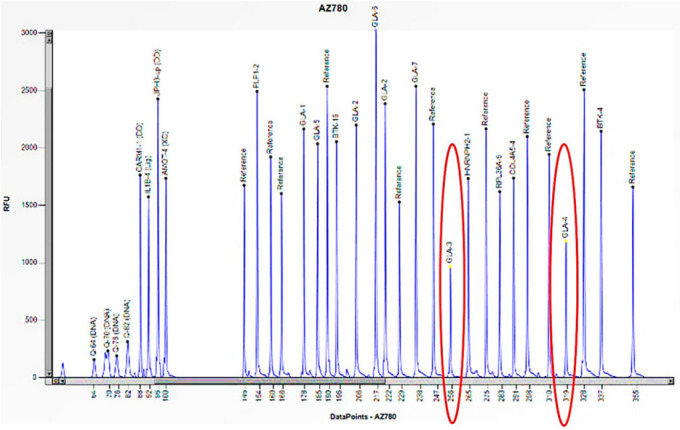
The electrophoretogram of the female patient with the code AZ780 demonstrates a 50% reduction in the peak corresponding to the probes for exons 3 and 4 of the *GLA* gene in comparison to the reference probe, thus indicating the presence of a heterozygous deletion.

**Figure 7 biomedicines-13-00973-f007:**
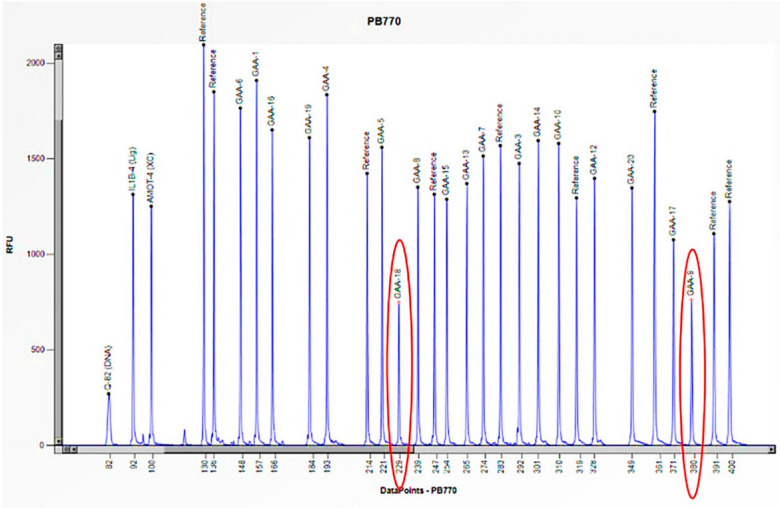
The electrophoretogram of the patient with the code PB770 demonstrates a 50% reduction in the peak corresponding to the probes for exons 9 and 18 of the *GAA* gene in comparison to the reference probe, thus indicating the presence of a heterozygous deletion.

**Figure 8 biomedicines-13-00973-f008:**
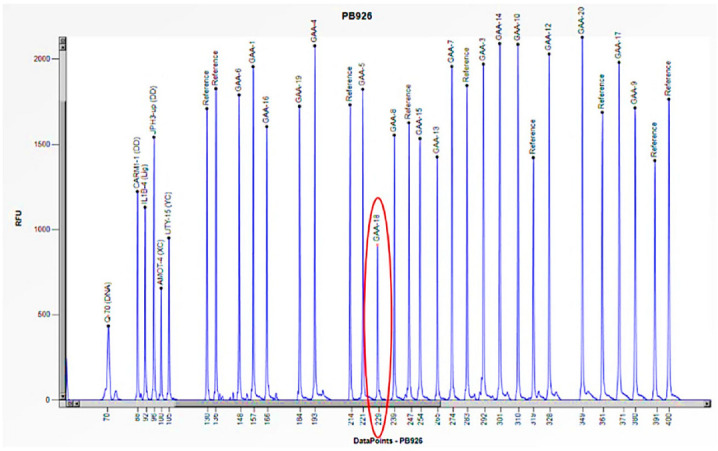
The electrophoretogram of the patient with the code PB926 demonstrates a 50% reduction in the peak corresponding to the probe of exon 18 of the *GAA* gene in comparison to the reference probe, thus indicating the presence of a heterozygous deletion.

**Figure 9 biomedicines-13-00973-f009:**
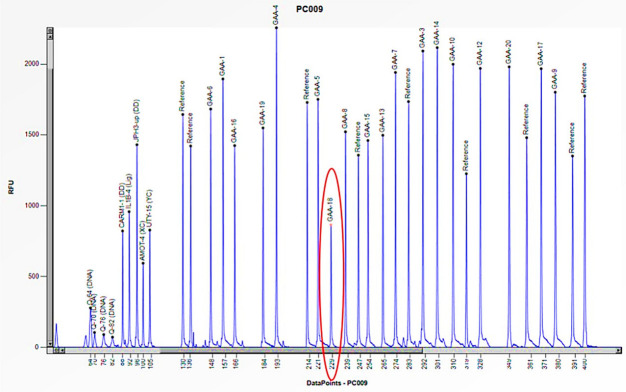
The electrophoretogram of the patient with the code PC009 demonstrates a 50% reduction in the peak corresponding to the probe of exon 18 of the *GAA* gene in comparison to the reference probe, thus indicating the presence of a heterozygous deletion.

**Figure 10 biomedicines-13-00973-f010:**
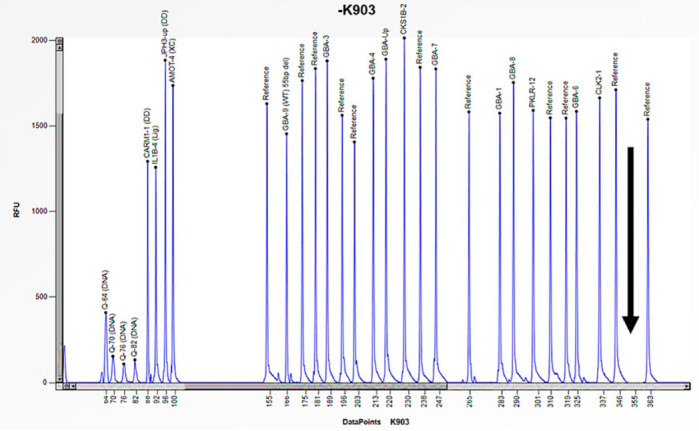
The electrophoretogram of the patient identified with the code K903 demonstrates a complete absence of the peak corresponding to the probe of exon 10 of the *GBA1* gene in comparison with the reference probe, thus indicating the presence of a homozygous deletion.

**Figure 11 biomedicines-13-00973-f011:**
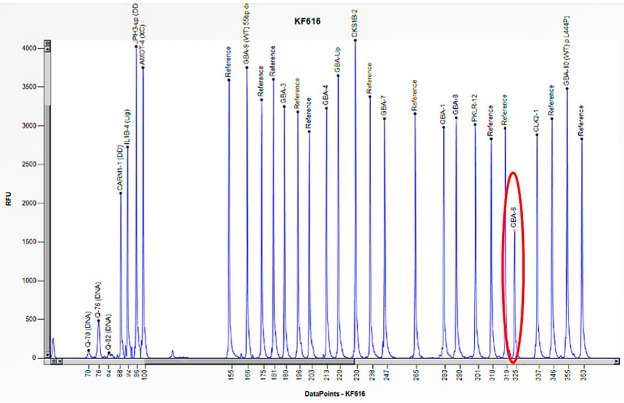
The electrophoretogram of the patient with the code KF616 demonstrates a 50% reduction in the peak corresponding to the probe of exon 6 of the *GBA1* gene in comparison to the reference probe, thus indicating the presence of a heterozygous deletion.

**Figure 12 biomedicines-13-00973-f012:**
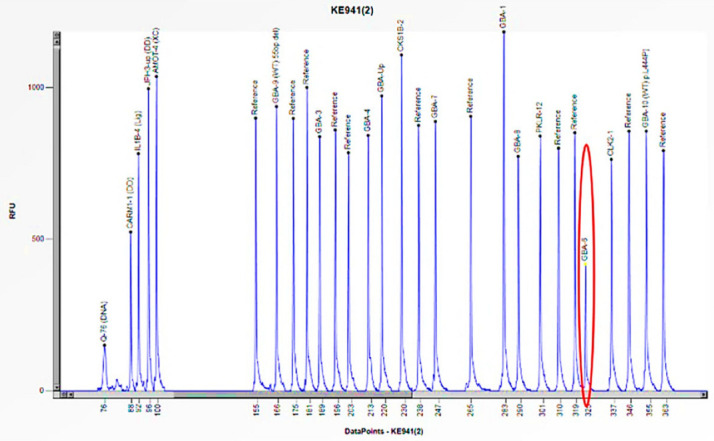
The electrophoretogram of the patient with the code KE941 demonstrates a 50% reduction in the peak corresponding to the probe of exon 6 of the *GBA1* gene in comparison to the reference probe, thus indicating the presence of a heterozygous deletion.

**Table 1 biomedicines-13-00973-t001:** Information on patients with Fabry disease and their relatives.

Family Member	Sex (F: Female, M: Male)	Age	Enzyme Activity (≥3 nmol/mL/h)	Lyso-Gb3 (0.08–1.13 nmol/L)	*GLA* Gene	Status
**Case 1 Family Study**						
1	M	32	0	36.64	Hemizygous deletion of exons 3 and 4	Acroparesthesias, cornea verticillata, angiokeratomas. Ischemic cardiomyopathy.
Mother	F	64	3.5	5.47	Heterozygous deletion of exons 3 and 4	Rheumatoid Arthritis
**Case 2 Family Study**						
2	M	46	0.2	35.37	Hemizygous deletion of exons 3 and 4	Left Ventricular Hypertrophy (LVH)
Mother	F	70	9.2	9.67	Heterozygous deletion of exons 3 and 4	Left Ventricular Hypertrophy (LVH)

**Table 2 biomedicines-13-00973-t002:** Information on patients with Pompe disease and their relatives.

Family Member	Sex (F: Female, M: Male)	Age	Enzyme Activity (≥6 nmol/mL/h)	*GAA* Gene	Status
3	F	1 month	0.9	E262K/Heterozygous deletion of exons 9 and 18	Severe cardiac involvement, generalized hypotonia, muscle weakness, proteinuria, and poor growth
Mother	F	37	5.9	Heterozygous deletion of exons 9 and 18	Asymptomatic
Father	M	42	5.0	E262K heterozygote	Asymptomatic
4	M	9	1.5	c.-32-13T > G/Heterozygous deletion of exon 18	Increased CPK, increased AST/ALT, increased LDH, slight neck weakness
5	M	24	1.2	c.-32-13T > G/Heterozygous deletion of exon 18	Hyperkemia, increased AST/ALT, muscle weakening, hepatomegaly. Muscle biopsy analysis (vacuolar myopathy with glycogen accumulation)

**Table 3 biomedicines-13-00973-t003:** Information on patients with Gaucher disease and their relatives.

Family Member	Sex (F: female, M: male)	Age	Enzyme Activity (≥2.5 nmol/mL/h)	Lyso-Gb1 (≤6.8 ng/mL)	*GBA1* Gene	Status
**Case 6 Family Study**						
6	M	2	0.4	--	Homozygous deletion of exon 10	Hepatosplenomegaly, anaemia, thrombocytopenia, and gastroesophageal reflux
Mother	F	38	3.5	--	Heterozygous deletion of exon 10	Asymptomatic
Father	M	43	4.5	--	Heterozygous deletion of exon 10	Asymptomatic
**Case 7 Family Study**						
7	F	22	0.5	498.0	N409S/Heterozygous deletion of exon 6	Hepatomegaly, splenomegaly, and thrombocytopenia
Mother	F	53	3.0	--	Heterozygous deletion of exon 6	Asymptomatic
Father	M	58	4.2	--	N409S heterozygote	Asymptomatic

## Data Availability

Data are contained within the article.

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
