# Peer review of "Multiplex Ligation Probe Amplification and Sanger Sequencing: Light and Shade in the Diagnosis of Lysosomal Storage Disorders"

_biomedicines, 2025, doi:10.3390/biomedicines13040973_

Round 1

Reviewer 1 Report

Comments and Suggestions for Authors

General Comments.  This manuscript uses MLPA to confirm the molecular genetic diagnosis of LSDs for 9 patients in whom no mutation (Fabry) or no or only one mutation (Gaucher and Pompe) were identified by Sanger sequencing. The manuscript goes back and forth between being educational and providing all of the raw data from each family study.  The manuscript could be significantly reduced in size by eliminating unnecessary data and reducing the number of figures.  This reviewer has several recommendations for strengthening the manuscript.

  1. In Section 1.1, it would be helpful to include an illustration of how the MLPA technique is performed and the bands quantified.
  2. In Section1.1, it would be informative to list the pros and cons of MLPA versus other molecular genetic techniques used to detect insertions and deletions (indels), e.g., next-generation sequencing (NGS) and chromosome microarray analysis (CMA), which are widely used for initial molecular genetic diagnosis in other settings. For example, MLPA is a simple, cost-efficient, and sensitive technique for detecting indels in specifically targeted genes.  The authors should also address how the potential for false positives and negatives with MLPA are minimized by selecting highly conserved and unique sequences and that complementary techniques, e.g., DNA sequencing, can be used to rule out the presence of a SNP that can prevent the binding of a probe to an exon sequence.  The authors fail to mention the possibility that there could be mutations in non-coding regions (promoters, introns, and 5’ and 3’ UTRs) or structural rearrangements (e.g., intra-genic inversions or translocations) that cannot be detected by their standard workflow of exon-based sequencing followed by MLPA, and which are best identified by genome sequencing.
  3. In Section 1.2, the authors should include the number of exons in each of the three genes and the number of MLPA probes used.
  4. In Section 2, the first paragraph is confusing.  The authors should clearly summarize how many of the 800 patients overall and for each disease had one (Fabry) or two (Gaucher and Pompe) mutations identified by Sanger sequencing and then by MLPA.  In Line 179, the authors state “These patients had previously undergone genetic analysis using Sanger's, which detected no mutations in specific genes (or only one causative mutation), despite typical symptoms and reduced or absent enzyme activity.” Does this mean that they confirmed a molecular genetic diagnosis in only 9 of 800 patients?  The numbers do not add up.
  5. The authors use various colloquial terms for Sanger sequencing (by Sanger, by Sanger’s, through Sanger, etc.). The authors should use the proper scientific term, Sanger sequencing, throughout the manuscript.
  6. The authors should italicize the abbreviated gene names throughout the manuscript as GLA, GBA1, and
  7. In Section, 2, the results for each case are redundant. This reviewer recommends showing the data one case as an example and then describing the remaining cases only in the text.

Specific Edits.

  1. Line 23. Change “reported” to “report”
  2. Line 24. Change “symptomatology” to “symptomatically”
  3. Line 25. Add “with” before “absent”
  4. Line 31. How did a male with a homozygous deletion of exon 10 not arise suspicion for a deletion by Sanger sequencing?  There would have been no amplicon to sequence, which the authors should point out, leading to the possibilities of a homozygous deletion or a homozygous SNP that prevented the sequencing primers from binding and amplifying.
  5. Line 54. Add “small deletions or insertions” after “point mutations”  Sequencing can detect indels within exons because the two sequences will be out of frame.
  6. Line 55. Change “deletions or insertions” to “large deletions or insertions spanning one or more exons”
  7. Line 61. “add” exon-spanning” before “deletions” because intra-exonic indels will be detected as two out-of-frame sequences by Sanger sequencing.
  8. Line 63. Change to “Multiplex Ligation-dependent Probe Amplification (MLPA)”
  9. Line 69. It would be helpful to the readers to add an illustration of how the MLPA technique works and how the results are analyzed.
  10. Line 73. Change “reaction” to “technique”
  11. Line 75. After “probes” add “with common primers”
  12. Line 76. Change “Data analysis” to “Quantification of amplification products.”
  13. Line 79. After ”sequences” add “with no nucleotides in between.”
  14. Line 80. Change “in” to “into”
  15. Line 80. After “amplification”, add “using primers that recognize common non-binding tail sequences at the 5’ ends of the probes.” 
  16. Lines 80-81. Change “The amplification reaction is performed using only one primer pair, one of which is fluorescently labelled.” to “Each amplification reaction is performed using two primers, only one of which is fluorescently labelled.”
  17. Line 83, 248, 277, 306, 347, 418, 456, 485, 564, 601, 636. Change “electropherogram” to “electrophoretogram”
  18. Line 85. Change “accumulated” to “amount”
  19. Line 93. Change “amplifications” to “duplications”
  20. Line 103. No need to capitalize “Lysosomal Storage Diseases” 
  21. Line 121. Italicize gene names, e.g., “GLA
  22. Line 121-122. Change “lysosomal enzyme in question” to “a-GAL A”
  23. Line 127. The authors are mixing up the concepts of mutation detection and X-inactivation.  Affected females are not reliably detected because they have two X chromosomes, not because one is inactive.  Change to “females, who despite having two X chromosomes, may develop an X-linked disease due to the phenomenon of lyonization (……”
  24. Lines 133-135. The first phrase seems out of place and should be the opening statement for the entire section. Alternatively, the authors could rectify this by including autosomal recessive inheritance in the first sentence, e.g., change “. These conditions” to “which”
  25. Line 137. Add “acid” before “b-glucosidase”
  26. Line 138. Add “of GCase activity” after “deficiency”
  27. Line 148 Add “the activity of” before “the acid alpha glucosidase”
  28. Line 153 Add “including hypertrophic cardiomyopathy” after “symptoms,”
  29. Lines 156-157. Change “In contrast to Fabry disease, which is associated with the X chromosome, Gaucher disease and Pompe disease are autosomal recessive disorders. The manifestation of the clinical phenotype in these diseases requires the mutation of both allelic copies, the maternal and paternal origin, respectively.” to “In contrast to Fabry disease, Gaucher and Pompe diseases are autosomal recessive disorders that require a mutation in both the maternal and paternal alleles.”
  30. Lines 163-164. Change “specific….heterozygosis” to “homozygous and heterozygous”
  31. Line 165. Why didn’t the homozygous deletion arouse suspicion by Sanger sequencing?
  32. Line 177. I am confused about the numbers.  Does 800 refer to patients with unidentified mutations or the total number of patients sequenced?  This number seems high for unexplained mutations, especially when the authors only present 9 cases in which an additional disease-causing mutation was identified.  The authors should define their dataset more clearly, i.e., the total number of patients sequenced for each disease and the number of patients with each disease in whom a disease-causing mutation was identified (one for Fabry and two for Gaucher and Pompe) or not identified (none for Fabry and one for Gaucher and Pompe).  There could still be patients with non-coding region mutations not be detected by Sanger sequencing or MLPA, e.g., deep intronic mutations, mutations in the 5’ or 3’ UTS, or promoter mutations, that can only be identified by genome sequencing.
  33. Line 192. Change “α-galactosidase A (α-GAL A)” to “α-GAL A”
  34. Line 192. Null typically refers to the absence of a protein, not the absence of enzymatic activity.  Change “a null result” to “little to no enzymatic activity”
  35. Line 194. There are two reasons why PCR products may not amplify: (1) there could be a nucleotide variant in the DNA sequence that prevents the primer from binding, and (2) the primer binding site and adjacent sequence may be deleted.  The authors should expand on these possibilities and how they determined that the amplicon failure was due to a deletion and not a failure of the PCR probe to bind.  For exon-based sequencing, the primers are located within introns, which could exhibit some sequence variation.
  36. Line 198. Change “the accumulation product” to “an accumulation substrate”
  37. Line 202. Change “with” to “had”
  38. Line 203. Delete “accumulation”
  39. Line 204. Change “threshold value” to “Lower limit of normal” or Reference value”
  40. Line 213. To reduce redundancy, the authors should combine the two Fabry family case studies into a single table with a single header for each column.  Each family case study should present the proband’s data followed by the mother’s findings to avoid jumping between sub-tables. Definitions of M and F can be removed from the table and placed as a caption.
  41. Line 213. “familiarity with ischemic cardiomyopathy” does not make sense in English.  Please correct the wording.
  42. Line 219. Each figure provides the quantification of the bands as well as the electrophoretogram for each case study.  This level of detail is totally unnecessary and extends the length of the manuscript considerably.  This reviewer recommends combining Figures 1 and 3 by showing the relevant electrophoretograms from the Family 1 proband as Part A and the mother as Part B, along with the bar graph as Part C. The differences between a total deletion, a heterozygous deletion, and reference are self-evident and do not require all of the quantitative values in a separate table and separate electrophoretograms.  The authors could choose to include the relevant quantitative data in the text to support the results of the peaks.  The authors should include the names of the two deleted bands GLA-3 and GLA-4 on top of the arrows in the electrophoretogram to guide the reader.
  43. Line 264. Combine Figures 2 and 4 and just show the electrophoretograms.
  44. Line 281. Figure 3 provides a lot of unnecessary details.  This reviewer recommends keeping the electrophoretogram but removing Parts A and C.  Furthermore, Part C is listed as FD #Case 1, but appears to refer to Case 2.
  45. Lines 313. This reviewer recommends just showing the bar graphs for each subsequent family study.  The amount of data in the current figures is unnecessary.
  46. Line 360. Change “analysis” to “activity”
  47. Line 361. Change “in” to “by”
  48. Line 362. Change “single causative mutation in heterozygosis” to single heterozygous mutation”  Mutations are, by definition, disease-causing.
  49. Line 362. Change “justifying” to “accounting for”
  50. Line 366. Change “deletion in heterozygosis” to “heterozygous deletion”
  51. Line 386. This reviewer recommends combining the Pompe disease cases into one table separated by Family Case Number, keeping the probands and parents together.
  52. Lines 388. Change to “E262K heterozygote”
  53. Line 368. Change “thanks to MLPA,” to “by MLPA”
  54. Lines 369, 371, 372, 378, 388, 502, 512, 516, 525, 682. Change “heterozygosis” to “heterozygosity”
  55. Lines 376, 508. Change “analysis” to “activity”
  56. Line 378. Delete “hetero”
  57. Line 384. Change “clinic” to “clinical features”
  58. Line 391. Just show the bar graphs for each family case study after the first Fabry family.
  59. Table 2. The fifth column is uninformative and should be deleted.
  60. Lines 213, 214, 528, 529, 530. In the second row, add Normal before the value
  61. Line 492. Use past tense for results, i.e., “exhibited” not “exhibits” and “were” instead of “are”
  62. Line 497. Change “justifying to “explaining”
  63. Line 497. I assume exon 10 did not amplify, which should have been suspicious for a deletion or nucleotide variant prevent the PCR primer from binding to the patient’s DNA
  64. Line 513. Change “also” to “historically”
  65. Line 517. Change “didn’t” to “did not”
  66. Line 517. Change “the deletion in heterozygosis” to “a heterozygous deletion”
  67. Line 528. Combine the family studies into one table with one header.  Keep the family members together.
  68. Line 530, 3rd Change to “N409S heterozygote”
  69. Line 538. Just keep the bar graphs for the Gaucher Family Studies.
  70. Line 649. After blood, add “urine, or tissue”
  71. Line 650. A small intragenic deletion or insertion show two out-of-frame sequences.  Only an entire exon deletion will yield a negative result.  Please re-write.
  72. Line 657. Change “distinct” to “different”
  73. Libe 662. Change “product” to “substrate”
  74. Line 679. Change “mutagenesis of both allelic copies (of paternal and maternal origin).” To “the presence of bi-allelic mutations”
  75. Line 703. Change “homology affinity” to “homology”
  76. Line 714. Change “homozygosis” to “apparent homozygosity”
  77. Line 729. Change “the clinical diagnosis” to “confirming the molecular genetic diagnosis”
Comments on the Quality of English Language

General Comments.  This manuscript uses MLPA to confirm the molecular genetic diagnosis of LSDs for 9 patients in whom no mutation (Fabry) or no or only one mutation (Gaucher and Pompe) were identified by Sanger sequencing. The manuscript goes back and forth between being educational and providing all of the raw data from each family study.  The manuscript could be significantly reduced in size by eliminating unnecessary data and reducing the number of figures.  This reviewer has several recommendations for strengthening the manuscript.

  1. In Section 1.1, it would be helpful to include an illustration of how the MLPA technique is performed and the bands quantified.
  2. In Section1.1, it would be informative to list the pros and cons of MLPA versus other molecular genetic techniques used to detect insertions and deletions (indels), e.g., next-generation sequencing (NGS) and chromosome microarray analysis (CMA), which are widely used for initial molecular genetic diagnosis in other settings. For example, MLPA is a simple, cost-efficient, and sensitive technique for detecting indels in specifically targeted genes.  The authors should also address how the potential for false positives and negatives with MLPA are minimized by selecting highly conserved and unique sequences and that complementary techniques, e.g., DNA sequencing, can be used to rule out the presence of a SNP that can prevent the binding of a probe to an exon sequence.  The authors fail to mention the possibility that there could be mutations in non-coding regions (promoters, introns, and 5’ and 3’ UTRs) or structural rearrangements (e.g., intra-genic inversions or translocations) that cannot be detected by their standard workflow of exon-based sequencing followed by MLPA, and which are best identified by genome sequencing.
  3. In Section 1.2, the authors should include the number of exons in each of the three genes and the number of MLPA probes used.
  4. In Section 2, the first paragraph is confusing.  The authors should clearly summarize how many of the 800 patients overall and for each disease had one (Fabry) or two (Gaucher and Pompe) mutations identified by Sanger sequencing and then by MLPA.  In Line 179, the authors state “These patients had previously undergone genetic analysis using Sanger's, which detected no mutations in specific genes (or only one causative mutation), despite typical symptoms and reduced or absent enzyme activity.” Does this mean that they confirmed a molecular genetic diagnosis in only 9 of 800 patients?  The numbers do not add up.
  5. The authors use various colloquial terms for Sanger sequencing (by Sanger, by Sanger’s, through Sanger, etc.). The authors should use the proper scientific term, Sanger sequencing, throughout the manuscript.
  6. The authors should italicize the abbreviated gene names throughout the manuscript as GLA, GBA1, and
  7. In Section, 2, the results for each case are redundant. This reviewer recommends showing the data one case as an example and then describing the remaining cases only in the text.

Specific Edits.

  1. Line 23. Change “reported” to “report”
  2. Line 24. Change “symptomatology” to “symptomatically”
  3. Line 25. Add “with” before “absent”
  4. Line 31. How did a male with a homozygous deletion of exon 10 not arise suspicion for a deletion by Sanger sequencing?  There would have been no amplicon to sequence, which the authors should point out, leading to the possibilities of a homozygous deletion or a homozygous SNP that prevented the sequencing primers from binding and amplifying.
  5. Line 54. Add “small deletions or insertions” after “point mutations”  Sequencing can detect indels within exons because the two sequences will be out of frame.
  6. Line 55. Change “deletions or insertions” to “large deletions or insertions spanning one or more exons”
  7. Line 61. “add” exon-spanning” before “deletions” because intra-exonic indels will be detected as two out-of-frame sequences by Sanger sequencing.
  8. Line 63. Change to “Multiplex Ligation-dependent Probe Amplification (MLPA)”
  9. Line 69. It would be helpful to the readers to add an illustration of how the MLPA technique works and how the results are analyzed.
  10. Line 73. Change “reaction” to “technique”
  11. Line 75. After “probes” add “with common primers”
  12. Line 76. Change “Data analysis” to “Quantification of amplification products.”
  13. Line 79. After ”sequences” add “with no nucleotides in between.”
  14. Line 80. Change “in” to “into”
  15. Line 80. After “amplification”, add “using primers that recognize common non-binding tail sequences at the 5’ ends of the probes.” 
  16. Lines 80-81. Change “The amplification reaction is performed using only one primer pair, one of which is fluorescently labelled.” to “Each amplification reaction is performed using two primers, only one of which is fluorescently labelled.”
  17. Line 83, 248, 277, 306, 347, 418, 456, 485, 564, 601, 636. Change “electropherogram” to “electrophoretogram”
  18. Line 85. Change “accumulated” to “amount”
  19. Line 93. Change “amplifications” to “duplications”
  20. Line 103. No need to capitalize “Lysosomal Storage Diseases” 
  21. Line 121. Italicize gene names, e.g., “GLA
  22. Line 121-122. Change “lysosomal enzyme in question” to “a-GAL A”
  23. Line 127. The authors are mixing up the concepts of mutation detection and X-inactivation.  Affected females are not reliably detected because they have two X chromosomes, not because one is inactive.  Change to “females, who despite having two X chromosomes, may develop an X-linked disease due to the phenomenon of lyonization (……”
  24. Lines 133-135. The first phrase seems out of place and should be the opening statement for the entire section. Alternatively, the authors could rectify this by including autosomal recessive inheritance in the first sentence, e.g., change “. These conditions” to “which”
  25. Line 137. Add “acid” before “b-glucosidase”
  26. Line 138. Add “of GCase activity” after “deficiency”
  27. Line 148 Add “the activity of” before “the acid alpha glucosidase”
  28. Line 153 Add “including hypertrophic cardiomyopathy” after “symptoms,”
  29. Lines 156-157. Change “In contrast to Fabry disease, which is associated with the X chromosome, Gaucher disease and Pompe disease are autosomal recessive disorders. The manifestation of the clinical phenotype in these diseases requires the mutation of both allelic copies, the maternal and paternal origin, respectively.” to “In contrast to Fabry disease, Gaucher and Pompe diseases are autosomal recessive disorders that require a mutation in both the maternal and paternal alleles.”
  30. Lines 163-164. Change “specific….heterozygosis” to “homozygous and heterozygous”
  31. Line 165. Why didn’t the homozygous deletion arouse suspicion by Sanger sequencing?
  32. Line 177. I am confused about the numbers.  Does 800 refer to patients with unidentified mutations or the total number of patients sequenced?  This number seems high for unexplained mutations, especially when the authors only present 9 cases in which an additional disease-causing mutation was identified.  The authors should define their dataset more clearly, i.e., the total number of patients sequenced for each disease and the number of patients with each disease in whom a disease-causing mutation was identified (one for Fabry and two for Gaucher and Pompe) or not identified (none for Fabry and one for Gaucher and Pompe).  There could still be patients with non-coding region mutations not be detected by Sanger sequencing or MLPA, e.g., deep intronic mutations, mutations in the 5’ or 3’ UTS, or promoter mutations, that can only be identified by genome sequencing.
  33. Line 192. Change “α-galactosidase A (α-GAL A)” to “α-GAL A”
  34. Line 192. Null typically refers to the absence of a protein, not the absence of enzymatic activity.  Change “a null result” to “little to no enzymatic activity”
  35. Line 194. There are two reasons why PCR products may not amplify: (1) there could be a nucleotide variant in the DNA sequence that prevents the primer from binding, and (2) the primer binding site and adjacent sequence may be deleted.  The authors should expand on these possibilities and how they determined that the amplicon failure was due to a deletion and not a failure of the PCR probe to bind.  For exon-based sequencing, the primers are located within introns, which could exhibit some sequence variation.
  36. Line 198. Change “the accumulation product” to “an accumulation substrate”
  37. Line 202. Change “with” to “had”
  38. Line 203. Delete “accumulation”
  39. Line 204. Change “threshold value” to “Lower limit of normal” or Reference value”
  40. Line 213. To reduce redundancy, the authors should combine the two Fabry family case studies into a single table with a single header for each column.  Each family case study should present the proband’s data followed by the mother’s findings to avoid jumping between sub-tables. Definitions of M and F can be removed from the table and placed as a caption.
  41. Line 213. “familiarity with ischemic cardiomyopathy” does not make sense in English.  Please correct the wording.
  42. Line 219. Each figure provides the quantification of the bands as well as the electrophoretogram for each case study.  This level of detail is totally unnecessary and extends the length of the manuscript considerably.  This reviewer recommends combining Figures 1 and 3 by showing the relevant electrophoretograms from the Family 1 proband as Part A and the mother as Part B, along with the bar graph as Part C. The differences between a total deletion, a heterozygous deletion, and reference are self-evident and do not require all of the quantitative values in a separate table and separate electrophoretograms.  The authors could choose to include the relevant quantitative data in the text to support the results of the peaks.  The authors should include the names of the two deleted bands GLA-3 and GLA-4 on top of the arrows in the electrophoretogram to guide the reader.
  43. Line 264. Combine Figures 2 and 4 and just show the electrophoretograms.
  44. Line 281. Figure 3 provides a lot of unnecessary details.  This reviewer recommends keeping the electrophoretogram but removing Parts A and C.  Furthermore, Part C is listed as FD #Case 1, but appears to refer to Case 2.
  45. Lines 313. This reviewer recommends just showing the bar graphs for each subsequent family study.  The amount of data in the current figures is unnecessary.
  46. Line 360. Change “analysis” to “activity”
  47. Line 361. Change “in” to “by”
  48. Line 362. Change “single causative mutation in heterozygosis” to single heterozygous mutation”  Mutations are, by definition, disease-causing.
  49. Line 362. Change “justifying” to “accounting for”
  50. Line 366. Change “deletion in heterozygosis” to “heterozygous deletion”
  51. Line 386. This reviewer recommends combining the Pompe disease cases into one table separated by Family Case Number, keeping the probands and parents together.
  52. Lines 388. Change to “E262K heterozygote”
  53. Line 368. Change “thanks to MLPA,” to “by MLPA”
  54. Lines 369, 371, 372, 378, 388, 502, 512, 516, 525, 682. Change “heterozygosis” to “heterozygosity”
  55. Lines 376, 508. Change “analysis” to “activity”
  56. Line 378. Delete “hetero”
  57. Line 384. Change “clinic” to “clinical features”
  58. Line 391. Just show the bar graphs for each family case study after the first Fabry family.
  59. Table 2. The fifth column is uninformative and should be deleted.
  60. Lines 213, 214, 528, 529, 530. In the second row, add Normal before the value
  61. Line 492. Use past tense for results, i.e., “exhibited” not “exhibits” and “were” instead of “are”
  62. Line 497. Change “justifying to “explaining”
  63. Line 497. I assume exon 10 did not amplify, which should have been suspicious for a deletion or nucleotide variant prevent the PCR primer from binding to the patient’s DNA
  64. Line 513. Change “also” to “historically”
  65. Line 517. Change “didn’t” to “did not”
  66. Line 517. Change “the deletion in heterozygosis” to “a heterozygous deletion”
  67. Line 528. Combine the family studies into one table with one header.  Keep the family members together.
  68. Line 530, 3rd Change to “N409S heterozygote”
  69. Line 538. Just keep the bar graphs for the Gaucher Family Studies.
  70. Line 649. After blood, add “urine, or tissue”
  71. Line 650. A small intragenic deletion or insertion show two out-of-frame sequences.  Only an entire exon deletion will yield a negative result.  Please re-write.
  72. Line 657. Change “distinct” to “different”
  73. Libe 662. Change “product” to “substrate”
  74. Line 679. Change “mutagenesis of both allelic copies (of paternal and maternal origin).” To “the presence of bi-allelic mutations”
  75. Line 703. Change “homology affinity” to “homology”
  76. Line 714. Change “homozygosis” to “apparent homozygosity”
  77. Line 729. Change “the clinical diagnosis” to “confirming the molecular genetic diagnosis”

Reviewer 2 Report

Comments and Suggestions for Authors

This study compares Multiplex Ligation Probe Amplification (MLPA), to conventional Sanger sequencing, for diagnosis of lysosomal diseases.   A weakness of conventional Sanger sequencing may have trouble identifying larger insertions and deletions and copy number variations (CNV). MLPA, however, can detect copy number variation (CNV), insertions/deletions, and greater than 40 DNA sequences in a single reaction.

In this study, 800 patients diagnosed with a probably lysosomal disease based on enzyme assay.  For these 800 patients, the diagnosis was not able to be confirmed using Sanger sequencing.  MLPA technique was applied to these 409 patient samples, and the result was that a molecular diagnostic mutation analysis was achieved in  9 patient cases (involving patients with Fabry disease, Pompe disease and Gaucher disease).

The authors give an excellent background/introduction description of Pompe disease, Fabry disease, Gaucher disease, as well as the MLPA process and key differences between MLPA and Sanger sequencing.

The authors conclude that MLPA should be a standard process included in the molecular diagnostic testing for lysosomal diseases.

It would be helpful if the authors could briefly describe what additional steps in general might need to be taken to identify mutation in patients with suspected lysosomal diseases for which Sanger sequencing and MLPA were unsuccessful in identifying the mutation.

Reviewer 3 Report

Comments and Suggestions for Authors

This study highlights the importance of MLPA in diagnosing Lysosomal Storage Diseases (LSDs), particularly when Sanger sequencing fails to detect large deletions or insertions. By analyzing nine patients with Fabry, Gaucher, and Pompe diseases, MLPA successfully identified mutations that traditional methods had missed. These findings suggest that MLPA should complement Sanger sequencing to ensure accurate and timely diagnoses, minimizing potential delays that could be critical for patients.

Suggested Revisions

  1. Lines 24–26 and 32–33: The section discussing MLPA’s high sensitivity in detecting larger CNVs needs clearer presentation. While MLPA effectively identifies large deletions or duplications, Sanger sequencing excels at detecting small-scale genetic variations such as single nucleotide variants (SNVs) and small insertions or deletions. However, due to its limitations in assessing copy number variations (CNVs), Sanger sequencing is not well-suited for detecting larger genomic alterations, such as exon-level deletions or duplications. This part of the abstract should be rewritten for improved clarity and logical flow.
  2. Lines 113–115: The paragraphs under the subheading “Principles of the MLPA Technique” do not align with the section’s focus. They should be moved to a more appropriate section, such as “1.2. MLPA in the Diagnosis of Fabry, Gaucher, and Pompe Diseases”.
  3. Subheading 1.2: MLPA in the Diagnosis of Fabry, Gaucher, and Pompe Diseases: This section provides general information about these diseases but lacks a structured discussion on MLPA’s role in their diagnosis. The content should be reorganized to focus on how MLPA enhances diagnostic accuracy for these diseases. A concluding paragraph should be added to highlight MLPA’s benefits in providing a more reliable diagnosis.
  4. Discussion Section: A major issue with this section is the lack of an in-depth discussion of the study’s findings. A more detailed analysis should be incorporated to contextualize the results, compare them with previous studies, and discuss their implications for clinical practice.
  5. Lines 733–738: This section lacks sufficient patient details, such as age, sex, and categorization. Including this information will improve the comprehensiveness of the study and provide better insight into patient demographics.

Round 2

Reviewer 1 Report

Comments and Suggestions for Authors

The revised manuscript is much improved, clearer to read, and has no duplication of results compared to the original manuscript.  However, there is still additional editing to be done as some of the new text is difficult to understand, not fully informative, or not grammatically correct.  For example, it is still unclear to this reviewer how many of the 800 patients had a confirmed biochemical diagnosis of Fabry, Gaucher, or Pompe disease.  Here are the specific comments and recommendations to further improve the manuscript:

  1. Line 30. Recommend changing the sentence to the following for clarity:  “Nine patients with deficient or absent enzyme activity had Sanger sequencing results that could not confirm the molecular genetic diagnosis because either no mutation (Fabry) or only one mutation (Gaucher and Pompe) was identified.
  2. Line 31. Remove “Only”
  3. Line 33. Change “with” to “had a”
  4. Line 34. In both instances, change “with” to “had a”
  5. Line 35. It is unclear who the remaining patients are since the authors describe the 9 patients with deletions.  Was MLPA performed on the other 791 as well or just in those with Sanger sequencing results that could not fully explain the enzyme deficiency?
  6. Line 96. Change “duplication product” to “amplicon”
  7. Line 98. Change “amplifications” to “duplications”
  8. Line 108. Change to: MLPA is a simple and innovative technique for determining the number of copies of a DNA sequence.  Its advantages include: 1) ability to scan up to 40 loci per gene; 2) wide range of applications; 3) high reproducibility and ease of execution; 4) ability to discriminate sequences differing by a single nucleotide; and 5) ability to discriminate small variations in copy number (3 versus 2) of a gene in a complex mixture [3]. Since MLPA is highly sensitive to the type of sample used for DNA extraction (blood or DBS), it is recommended that comparative analyses be performed using DNA samples extracted from the same tissue and by the same method. MLPA and Sanger sequencing focused on the coding sequence (exons) may fail to detect certain rare mutations present in the regulatory regions of a gene or that are caused by a structural rearrangement, e.g. an inversion or  balanced translocation. In these cases, genome sequencing, gene expression, fluorescence in situ hybridization, or other methods, may be required.”
  9. Line 132. Change to “A difference in peak height or peak area relative to a control”
  10. Line 143. add “easily” before “identify”
  11. Line 145 change “the double copy” to “two gene copies”
  12. Line 145 add “encompassing one or more exons” after “deletion”
  13. Line 148. Change “allelic” to “orthogonal”
  14. Line 160 change “a 95% homology affinity” to “95% homology”
  15. Line 172: change to “characterized by severe muscle weakness and hypertrophic cardiomyopathy, and the non-classic late-onset form (LOPD), which is characterized by slowly progressive muscle weakness.”
  16. Line 180, after “interest” add “for copy number variations”
  17. Line 237 change to “This study of 800 patients included 409 with a clinical
    suspicion of Fabry Disease (FD), 233 with a clinical suspicion of Gaucher Disease (GD) and 158 with a clinical suspicion of Pompe Disease (PD).”
  18. Line 241. It is still unclear how many of the 800 patients had deficient or absent enzymatic activity confirming a biochemical diagnosis of Fabry, Gaucher, or Pompe disease. As the authors state earlier in the manuscript, females with suspected Fabry disease require molecular genetic testing for disease confirmation because enzyme activity is often normal.  A flow diagram beginning with the 800 patients at the top, then segregated in to the three disease groups, then followed by enzyme activity results, then followed by Sanger sequencing results, and finally followed by MLPA results could be useful to clarify which evaluations each patient underwent.
  19. Line 274. Table 1.  Please align the columns for Cases 1 and 2.  Change “Case” to “Family member” in the first column-first row. 
  20. Line 266. Change “indicative” to “informative”
  21. Line 296 and 307. Add “male” before “patient”
  22. Line 318. Add “female” before “patient”
  23. Line 337 Change “were subjected to” to “had”
  24. Line 341 change “life,” to “life with”
  25. Line 344 change “allowed the identification of” to “only identified”
  26. Line 357 change “hyperkemia” to hyperCKemia”
  27. Line 358 change “hyposthenia” to “weakness”
  28. Line 361 change “justifying” to “explaining the”
  29. Line 367 change “in a biallelic condition” to “biallelic”
  30. Line 369. Delete column 5 as there is no plasma biomarker data presented. In Case 5, glycogen accumulation is already mentioned in the last column so it is redundant.
  31. Line 370. Table 2.  Change “hyposthenia” to “weakness”
  32. Line 471. The statement seems misleading as I would have thought that Sanger sequencing would have shown that exon 10 failed to amplify and could not be sequenced. If this is the case, it would be more accurate to state that no mutations were detected by Sanger sequencing, although exon 10 failed to amplify.  On the other hand, if the GAA gene rearrangement with its pseudogene somehow allowed for exon 10 to amplify in Sanger sequencing, this should be explained in Line 426.
  33. Line 430 change “However” to “In addition”
  34. Line 439 change “the presence of mutations” to “a mutation”
  35. Line 450. Table 3 is missing a header for Lyso-GL1 and its reference range
  36. Line 532 change “ ca cellular mosaic” to “cellular mosaicism”
  37. Line 533 add ”a mixed population of” before “cells”
  38. Line 35 change “healthy to “normal”
  39. Line 590 change to “The present study confirms that in subjects with large deletions, a molecular genetic diagnosis of an LSD may not be possible if only the Sanger sequencing method is used.”

Line 592.  Recommend changing to: “ Consequently, we propose that in cases of LSDs characterized by autosomal recessive transmission, MLPA should be conducted in all instances where the presence of clinical symptoms and no or significantly reduced enzyme activity are not fully explained by Sanger sequencing, or where familial testing does not align with the results obtained by Sanger sequencing, e.g. in the case of apparently homozygous mutation being due to a heterozygous deletion and a second mutation. In cases of LSDs characterized by X-linked transmission, MLPA should be performed in all suspected females as a first line test, as biochemical testing may be normal due to lyonization. In males with pathological enzyme activity and clinical symptoms of an X-linked disorder and uninformative Sanger sequencing results, MLPA is essential to detect the presence of a large deletion or insertion.  In conclusion, our data suggest that the MLPA technique may improve the genetic analysis of LSDs, limiting possible delays in molecular genetic diagnosis.”

Line 602.  A patient would still be treated based on clinical suspicion and enzyme deficiency.

Comments on the Quality of English Language

The revised manuscript is much improved, clearer to read, and has no duplication of results compared to the original manuscript.  However, there is still additional editing to be done as some of the new text is difficult to understand, not fully informative, or not grammatically correct.  For example, it is still unclear to this reviewer how many of the 800 patients had a confirmed biochemical diagnosis of Fabry, Gaucher, or Pompe disease.  Here are the specific comments and recommendations to further improve the manuscript:

  1. Line 30. Recommend changing the sentence to the following for clarity:  “Nine patients with deficient or absent enzyme activity had Sanger sequencing results that could not confirm the molecular genetic diagnosis because either no mutation (Fabry) or only one mutation (Gaucher and Pompe) was identified.
  2. Line 31. Remove “Only”
  3. Line 33. Change “with” to “had a”
  4. Line 34. In both instances, change “with” to “had a”
  5. Line 35. It is unclear who the remaining patients are since the authors describe the 9 patients with deletions.  Was MLPA performed on the other 791 as well or just in those with Sanger sequencing results that could not fully explain the enzyme deficiency?
  6. Line 96. Change “duplication product” to “amplicon”
  7. Line 98. Change “amplifications” to “duplications”
  8. Line 108. Change to: MLPA is a simple and innovative technique for determining the number of copies of a DNA sequence.  Its advantages include: 1) ability to scan up to 40 loci per gene; 2) wide range of applications; 3) high reproducibility and ease of execution; 4) ability to discriminate sequences differing by a single nucleotide; and 5) ability to discriminate small variations in copy number (3 versus 2) of a gene in a complex mixture [3]. Since MLPA is highly sensitive to the type of sample used for DNA extraction (blood or DBS), it is recommended that comparative analyses be performed using DNA samples extracted from the same tissue and by the same method. MLPA and Sanger sequencing focused on the coding sequence (exons) may fail to detect certain rare mutations present in the regulatory regions of a gene or that are caused by a structural rearrangement, e.g. an inversion or  balanced translocation. In these cases, genome sequencing, gene expression, fluorescence in situ hybridization, or other methods, may be required.”
  9. Line 132. Change to “A difference in peak height or peak area relative to a control”
  10. Line 143. add “easily” before “identify”
  11. Line 145 change “the double copy” to “two gene copies”
  12. Line 145 add “encompassing one or more exons” after “deletion”
  13. Line 148. Change “allelic” to “orthogonal”
  14. Line 160 change “a 95% homology affinity” to “95% homology”
  15. Line 172: change to “characterized by severe muscle weakness and hypertrophic cardiomyopathy, and the non-classic late-onset form (LOPD), which is characterized by slowly progressive muscle weakness.”
  16. Line 180, after “interest” add “for copy number variations”
  17. Line 237 change to “This study of 800 patients included 409 with a clinical
    suspicion of Fabry Disease (FD), 233 with a clinical suspicion of Gaucher Disease (GD) and 158 with a clinical suspicion of Pompe Disease (PD).”
  18. Line 241. It is still unclear how many of the 800 patients had deficient or absent enzymatic activity confirming a biochemical diagnosis of Fabry, Gaucher, or Pompe disease. As the authors state earlier in the manuscript, females with suspected Fabry disease require molecular genetic testing for disease confirmation because enzyme activity is often normal.  A flow diagram beginning with the 800 patients at the top, then segregated in to the three disease groups, then followed by enzyme activity results, then followed by Sanger sequencing results, and finally followed by MLPA results could be useful to clarify which evaluations each patient underwent.
  19. Line 274. Table 1.  Please align the columns for Cases 1 and 2.  Change “Case” to “Family member” in the first column-first row. 
  20. Line 266. Change “indicative” to “informative”
  21. Line 296 and 307. Add “male” before “patient”
  22. Line 318. Add “female” before “patient”
  23. Line 337 Change “were subjected to” to “had”
  24. Line 341 change “life,” to “life with”
  25. Line 344 change “allowed the identification of” to “only identified”
  26. Line 357 change “hyperkemia” to hyperCKemia”
  27. Line 358 change “hyposthenia” to “weakness”
  28. Line 361 change “justifying” to “explaining the”
  29. Line 367 change “in a biallelic condition” to “biallelic”
  30. Line 369. Delete column 5 as there is no plasma biomarker data presented. In Case 5, glycogen accumulation is already mentioned in the last column so it is redundant.
  31. Line 370. Table 2.  Change “hyposthenia” to “weakness”
  32. Line 471. The statement seems misleading as I would have thought that Sanger sequencing would have shown that exon 10 failed to amplify and could not be sequenced. If this is the case, it would be more accurate to state that no mutations were detected by Sanger sequencing, although exon 10 failed to amplify.  On the other hand, if the GAA gene rearrangement with its pseudogene somehow allowed for exon 10 to amplify in Sanger sequencing, this should be explained in Line 426.
  33. Line 430 change “However” to “In addition”
  34. Line 439 change “the presence of mutations” to “a mutation”
  35. Line 450. Table 3 is missing a header for Lyso-GL1 and its reference range
  36. Line 532 change “ ca cellular mosaic” to “cellular mosaicism”
  37. Line 533 add ”a mixed population of” before “cells”
  38. Line 35 change “healthy to “normal”
  39. Line 590 change to “The present study confirms that in subjects with large deletions, a molecular genetic diagnosis of an LSD may not be possible if only the Sanger sequencing method is used.”

Line 592.  Recommend changing to: “ Consequently, we propose that in cases of LSDs characterized by autosomal recessive transmission, MLPA should be conducted in all instances where the presence of clinical symptoms and no or significantly reduced enzyme activity are not fully explained by Sanger sequencing, or where familial testing does not align with the results obtained by Sanger sequencing, e.g. in the case of apparently homozygous mutation being due to a heterozygous deletion and a second mutation. In cases of LSDs characterized by X-linked transmission, MLPA should be performed in all suspected females as a first line test, as biochemical testing may be normal due to lyonization. In males with pathological enzyme activity and clinical symptoms of an X-linked disorder and uninformative Sanger sequencing results, MLPA is essential to detect the presence of a large deletion or insertion.  In conclusion, our data suggest that the MLPA technique may improve the genetic analysis of LSDs, limiting possible delays in molecular genetic diagnosis.”

Line 602.  A patient would still be treated based on clinical suspicion and enzyme deficiency.

Reviewer 3 Report

Comments and Suggestions for Authors

The abstract contains an overly lengthy background.
